# An Exploratory Analysis of the Association between Healthcare Associated Infections & Hospital Financial Performance

**DOI:** 10.3390/healthcare12131314

**Published:** 2024-06-30

**Authors:** Brad Beauvais, Diane Dolezel, Ramalingam Shanmugam, Dan Wood, Rohit Pradhan

**Affiliations:** 1School of Health Administration, Texas State University, Encino Hall, Room 250A, 601 University Drive, San Marcos, TX 78666, USA; 2Health Informatics & Information Management Department, Texas State University, Round Rock, TX 78665, USA; 3Baylor University, Army-Baylor University MHA/MBA Program, U.S. Army Medical Center of Excellence, San Antonio, TX 78234, USA

**Keywords:** hospital, infections, patient safety, HAI, financial performance

## Abstract

**Purpose:** Healthcare-associated infections (HAIs) place a significant financial burden on United States hospitals. HAI treatments extend hospital lengths of stay and increase hospital operational costs while significantly reducing hospital profit margins. Given these challenges, the research aim of this study was to explore the association between HAIs and hospital financial performance. A better understanding of this relationship can assist hospital leaders in optimizing the use of scarce financial resources to reduce HAI prevalence. **Methods:** Data for calendar year 2022 for active short-term acute care hospitals (*n* = 1454) in the US were analyzed using multiple linear regression analysis. We explored two derived dependent variables, operating expense per staffed bed and operating expense per discharge. The independent variables included four healthcare-associated infection rates: methicillin-resistant *Staphylococcus aureus* (MRSA) infection rate, Clostridium difficile (C. *diff)* infection rate, Catheter-Associated Urinary Tract Infection (CAUTI) rate, and Central Line Associated Blood Stream Infections (CLABSI). Appropriate organizational and market-level variables that may independently influence hospital financial performance were included as control variables. **Results:** The results revealed that C. *diff* (*β*: 0.037, *p* < 0.05) and CAUTI (*β*: 0.031, *p* < 0.05) rates were positively associated with an increase in operating expense per staffed bed, while increases in MRSA (*β*: 0.042, *p* < 0.001), C. *diff* (*β*: 0.062, *p* < 0.001), and CAUTI rates (*β*: 0.039, *p* < 0.001) were correlated with increased operating expenses per discharge. **Conclusions:** This study demonstrates that specific HAIs may be associated with increased hospital expenses. Proactively targeting these infections through tailored interventions may lead to reduced hospital costs, improved financial performance, and economic stability.

## 1. Introduction

### Background

Hospitals have encountered various operational challenges that have significantly impacted overall profitability and the capacity to treat patients [1]. Numerous hospitals have not been able to sustain operations and have been forced to declare bankruptcy, with nearly 80 Chapter 11 bankruptcy filings in 2023 alone [2]. Beyond the provision of health-sustaining and lifesaving capabilities, the hospital industry embodies over ten percent of the United States (U.S.) workforce [3]. Thus, the financial sustainability of hospitals should be among the top priorities of policymakers at the local, state, and national levels. Without a firm economic footing, every hospital’s ability to address patients’ and the local community’s public health needs is negatively impacted. Yet, the average US hospital struggles to maintain profitability in a complex system where hospitals are funded via a mix of government, commercial, and individual patient’s reimbursements. In 2022, the median operating margin for U.S. hospitals stood at a negative 3.8%, while the average operating margin was negative 13.5% [4]. By December 2023, the median operating margin had improved to 2.3%, but it is still far from ideal and remains questionably sustainable, as the number of recent bankruptcy filings attests [4].

With these recognized financial challenges in the hospital setting, the need to provide safe, effective, and efficient care has never been more vital. Hospitals simply cannot afford to introduce additional costs into the care delivery process and must focus on removing waste and error in every episode of care. Central to this premise is the effort to reduce and eliminate healthcare-associated infections (HAIs). HAIs are infections acquired during the first 48 h of receiving healthcare treatment that were not present or incubating at the time of admission or treatment [5]. Common healthcare-associated infections tracked by the Centers for Disease Control and Prevention (CDC) include central line-associated bloodstream infections (CLABSI), catheter-associated urinary tract infections (CAUTI), surgical site infections (SSI), ventilator-associated pneumonia (VAP), Methicillin-resistant *Staphylococcus aureus* (MRSA) infections, and Clostridioides difficile (C. *diff*) infections [6,7].

Reducing the number of HAIs is a priority for the U.S. Department of Health and Human Services (HHS) [8]. HAIs are costlier to treat because many of the causative microorganisms are antibiotic-resistant, and because sepsis, a life-threatening response to infection with potential multi-organ failure, is a frequent HAI complication [9,10,11]. Also, HAIs frequently occur in Intensive Care Units (ICUs) where patients in debilitated states undergo invasive procedures.

To reduce the disease burden of HAIs, many strategies have been applied including increasing the usage of antimicrobials, personal protective equipment, and the application of disinfectants across the healthcare environment [12,13]. For CLABSI and CAUTI specifically, as they are related to devices entering the body, improper aseptic or sterile techniques for cleaning, catheter insertion, or catheter maintenance may lead to the development of these HAIs [14]. When these techniques are not performed properly for urinary catheter insertion, microbes may enter the urinary tract and cause infection (CAUTI) in a patient’s kidneys, ureters, or bladder [15]. Likewise, if the sterile technique is not followed when inserting a central line, microbes may enter the bloodstream through a major vein leading directly to the heart [16]. However, although MRSA and C. *diff* are also HAIs, their etiology is much different. Both MRSA and C. *diff* can be spread from person-to-person or even from touching contaminated surfaces or everyday objects; handwashing and disinfecting surfaces and objects will help reduce the risk of these HAIs from further spread [17,18].

At the national level in the United States, the estimated treatment costs for HAIs range from US $28.4 to US $45.0 billion annually [19]. A recent systematic review by Serra-Burriel et al. (2020) determined that HAIs were strongly associated with higher direct costs, longer average length of stay, and higher mortality rates [20]. An additional systematic review conducted by the U.S. Agency for Healthcare Research (AHRQ) places wide ranges on the cost of various HAI infections. According to this study, the cost to treat a CAUTI infection ranges between US $4.6 K to US $29.7 K, a CLABSI case might cost US $17.9 K–US $94.9 K, and a C. *diff* case might cost from US $4.2 K to US $32.4 K [21].

The Centers for Medicare and Medicaid Services (CMS) Hospital-Acquired Condition Reduction Program is part of the Medicare Value-based Purchasing Program (VBP), and it reduces payments to hospitals based upon their performance on HAIs. Furthermore, most insurers do not reimburse hospitals for the cost of HAI treatment [7,8]. These reimbursement policies provide an incentive for hospitals to decrease HAI infection rates and with recent attention, notably between 2021 and 2022, these efforts have led to a 9% decrease in CLABSIs, a 12% decrease in CAUTIs, 16% decrease in MRSA, and 3% decrease in C. *diff* infections [18]. However, HAIs continue to adversely impact the care of patients and the hospitals in which they are treated. Annually, about 1.7 million people, or roughly one in every 31 hospital patients in the United States, suffer from one or more HAIs, leading to an estimated 98,000 deaths [22]. Perhaps more worrisome is the assertion that HAIs will increase in prevalence due to the advancing age of the US population, coupled with increasing patient acuity, shortages of skilled nurses, and an increase in the number of multidrug-resistant organisms (MDROs) [18].

Internationally, politicians, public health leaders, and researchers have increased their focus on the importance of addressing HAIs citing issues with prolonged hospital stays, long-term disability, increased costs for health systems and families, and premature death [23]. In recent studies, researchers noted the importance of reducing HAIs to improve patient clinical outcomes and help manage overall treatment costs. The European Centre for Disease Prevention and Control reported 3.5 million cases of HAIs estimated in the European Nation and European Economic area, leading to 90 thousand HAI-associated deaths [24]. From a more granular perspective, a health economic impact study initiative in Brazil reported the associated cost savings of US $8480 for CLABSIs, US $10,039 for VAP cases, and US $7464 for CAUTIs when these infections are well managed and eliminated [25,26]. When extrapolated to the total number of HAIs estimated to have been prevented by the project, the authors noted an overall savings of US $68.8 million and an estimated return on investment of 765% [27,28].

Consequently, reducing and eliminating HAIs is not only the appropriate way to treat patients entrusted to hospital care, but it also makes good business sense. What is less well understood is which HAI disease processes are the most closely associated with hospital profitability. While it is logical to assume that the increased operational costs of HAI would impact hospital profitability, to our knowledge this has never been tested at the hospital level. Advancing an understanding of which HAIs are costlier and have a statistically significant relationship with overall profitability can assist hospital leaders in prioritizing clinical interventions that are both clinically appropriate and improve overall economic viability.

## 2. Methods

### 2.1. Aim of the Study & Hypotheses

Given the thoughts reflected above, the aim of this study was to examine four specific HAI disease processes and their associated impact on hospital costs based on recent short-term acute care hospital data. Our targeted diseases included the methicillin-resistant *Staphylococcus aureus* (MRSA) infection rate, the Clostridium difficile (C. *diff)* infection rate, the Catheter-Associated Urinary Tract Infection (CAUTI) rate in ICUs and other wards, and the Central Line Associated Blood Stream Infections (CLABSI) rate in ICUs and other wards. We focused our attention on these four HAI rates due to data missingness in other hospital acquired conditions in our national sample of short-term acute care hospitals.

Based on the body of research on HAIs we conjectured that HAIs are likely to be associated with reduced financial outcomes for hospitals. Therefore, we hypothesize that increases in the targeted HAI infection rates (MRSA, C. *diff,* CAUTI, CLABSI) are significantly associated with worse financial performance in the short-term acute care hospital setting.

### 2.2. Data

Data were obtained from Definitive Healthcare which consolidates information from numerous public access databases pertaining to hospitals in the United States, such as the American Hospital Association Annual Survey (hospital profile), Medicare Cost Report (financial data), and the Hospital VBP (quality data) [29]. The original data set consisted of all 3876 short-term acute care hospitals in the United States. All Federal hospitals, including 172 Veterans Affairs, 26 Indian Health Service, and 31 Military Health System facilities, were excluded from our study sample due to a lack of numerous relevant data elements. We removed an additional 2193 facilities due to significant data missingness in our dependent, independent, and control variables considered in our analysis. The final dataset encompassed 1454 hospital observations from 2022, or roughly 38% of the total active short-term acute care facility population in the United States.

### 2.3. Dependent Variable

To evaluate the potential financial impact of healthcare-acquired infections in the short-term acute care hospital setting, we focused our attention on two dependent derived variables, operating expense per staffed bed, and operating expense per discharge, which are related to operating expenses for the year 2022. The year 2022 was specifically chosen as it was the most recent complete year of data. Hospital operating costs are a portion of operating expenses specifically related to patient care. In our initial plans, we also considered operating income as a possible dependent variable, however, as noted previously, hospitals generally will not be able to generate income directly because HAIs are not reimbursed by insurance companies. Thus, we narrowed our focus to our current variables related to operating expenses.

The first dependent variable we derived is “operating expense per staffed bed.” Hospital operating costs per staffed bed is a calculation that divides hospital operating costs associated with patient care (e.g., labor, supplies, etc.) by the number of staffed (vs. licensed) beds in the facility. We considered this variable to ascertain the relative overall operational costs for each hospital on an equitable basis as costs are distributed on a per-bed basis. In our opinion, this variable provides a perspective of the relative scale of expenses involved with health care delivery and is automatically scaled for size—making relative analysis and inferences across different size organizations logical and meaningful.

The second dependent variable we utilized is the “operating expense per adjusted discharge”. This measure is like our first dependent variable, but instead of distributing operational costs across only inpatient beds, we allocate costs across the number of adjusted discharges completed during the reporting period. A “discharge” is the release of a patient from care, but an “adjusted discharge” includes an adjustment to account for both inpatient and outpatient care volume. In our opinion, this variable provides a different view of operational performance, as the fixed number of bed constraints is replaced with a measure of total facility output. The calculation for an ‘adjusted discharge’ is reflected as:Adjusted Discharges = Inpatient Discharges + [(Gross Outpatient Revenue/Gross Inpatient Revenue) × Inpatient Discharges]

### 2.4. Independent Variables of Interest

Our study considered four types of HAIs as our independent variables of interest. These included the MRSA infection rate, C. *diff* infection rate, the CAUTI rate in ICUs and other wards, and the CLABSI rate in ICUs and other wards. These rates are reported as Standardized Infection Ratios (SIRs) which compare the actual number of HAIs reported vs. what would be predicted, given the standard population (i.e., National Healthcare Safety Network baseline) and adjusting for risk factors found to be significantly associated with differences in infection incidence [25]. HAI measures apply to all patients unless they fall into an excluded category as defined by CMS. Specifically, CLABSI and CAUTI data include infections that occurred in patients in ICUs, neonatal ICUs (for CLABSI only), and medical, surgical, and medical/surgical ward locations. MRSA bacteremia and C. *diff* data include those infections identified in all patients within the hospital [25].

### 2.5. Controls

Numerous independent variables are included in the study to account for the variation in operating costs associated with various individual hospital and hospital market characteristics, including the level of contract labor utilization (in millions of dollars), whether the facility is managed or owned, sole community hospital status, Accountable Care Organization (ACO) affiliation, the labor compensation ratio, market concentration (as measured via the Herfindahl–Hirschman Index (HHI))), the number of full time equivalent employees, government operated (or not), for-profit ownership (or not), academic medical center designation, urban versus rural location, the level of uncompensated care (in millions), average length of stay, Medicaid days of service, Medicare days of service, patient acuity (overall case mix index, the complication/comorbid and major complication/comorbid (CC/MCC) rate), the Hospital VBP Total Performance Score from the prior year (quality), and the geographic region of the state as defined by the American Hospital Association (Regions 1–9).

### 2.6. Analysis

Shapiro-Wilke normality tests were performed on both dependent variables and non-normality was observed (*p* < 0.001). In response, both dependent variables were shifted via min-max scaling and natural log transformations. To aid with ease of interpretation, the independent variables of interest were also natural log transformed to create a log-log model, where log Y_i_ = α + β log X_i_ + ε_i_. In instances where both the dependent variable and independent variable(s) are log-transformed variables, results may be interpreted as an expected percentage change in Y when X increases by some percentage. Such relationships, where both Y and X are log-transformed, are commonly referred to as elastic relationships in econometrics [30].

Eight multiple linear regressions with listwise deletion were conducted using IBM (International Business Machines) SPSS (Statistical Package for Social Sciences) Statistics package 28 [31]. In each of the analyses performed, the association between the studied independent variables and the dependent variable was rejected at an α = 0.05. Model fit was assessed using adjusted *R*^2^. Multicollinearity was evaluated and all variables maintained a variance inflation factor under 5 in all analyses.

## 3. Results

A descriptive analysis of all variables is available in Table 1. Our sample is comprised of 9% academic medical centers (SD = 0.28), 8% sole community hospitals (SD = 0.27), 8% rural hospitals (SD = 0.27), 11% are government-operated (SD = 0.32), 17% are for-profit (SD = 0.38), and the highest percentage of facilities are located in AHA Region 4 which includes the states of AL, FL, GA, MS, SC, TN and PR (17%; SD = 0.38). On average, our sample hospitals employ 1960 full-time equivalent staff (SD = 1688.74), spend US $20 million on contract staff (SD = US $21.88), spend US $34.8 million on uncompensated care (SD = US $50.65), maintain a case mix index of 1.85 (SD = 0.27), experience a CC/MCC rate of 69% (SD = 0.05), manage an average length of stay of 5.14 days (SD = 0.88), and 71% are affiliated with at least one ACO (SD = 0.46). On average, these facilities’ patients are comprised of 8% Medicaid (SD = 0.08%) and 26% Medicare (SD = 0.09%).

In our regression models, we tested all four HAI disease processes concurrently as our independent variables and regressed them on our two dependent variables pertaining to operating expense. In general, we observed statistically significant relationships in five of the eight tested models with directional consistency related to our independent variables of interest. Table 2 presents the multivariable regression results for our first four analyses related to operating expenses per staffed bed. The beta coefficients, standard error (S.E.), and significance (Sig.) are given for the natural log of operating expense per staffed bed. Our regression findings indicated that infection rates were associated with higher levels of expense across two of the first four infection rates tested. In the first set of analyses, C. *diff* infection was found to be statistically associated with operating expense per staffed bed (*R*^2^ = 48.2%, *β*: 0.037, S.E.: 0.014, *p* < 0.05). Similar findings were observed in the analysis of the CAUTI infections (*R*^2^ = 48.0%, *β*: 0.031, S.E.: 0.014, *p* < 0.05). The MRSA and CLABSI independent variables were not found to be significant in this analysis.

Table 3 presents multivariable regression results for our second dependent variable related to operating expense per adjusted discharge. Once again, we tested the natural log of the four HAI independent variables of interest regressed against the natural log of operating expense per adjusted discharge. In this set of variables, our findings still generally indicated that HAI infections are associated with higher levels of operating expense, with one exception: CLABSI infections were not significant. However, operating expense per adjusted discharge was positively associated with MRSA infection (*R*^2^ = 58.0%, *β*: 0.042, S.E.: 0.013, *p* < 0.001), C. *diff* infection (*R*^2^ = 58.6%, *β*: 0.062, S.E.: 0.011, *p* < 0.001), and CAUTI infection (*R*^2^ = 58.0%, *β*: 0.039, S.E.: 0.011, *p* < 0.001).

## 4. Discussion

According to the World Health Association (WHO), HAIs are the most frequently occurring adverse events in healthcare. However, they also suggest that HAI infection rates may be underreported [23]. Globally, there are challenges with obtaining reliable data due to the complexity of reporting HAIs and the lack of availability of HAI reporting systems. In Ethiopia, the second most populus country in Africa, there is not a reliable national HAI surveillance reporting system [32]. Even within the United States, reporting HAIs is not mandatory for all states or all types of hospitals. Despite these reporting challenges, those cases that are reported across the globe are enormous in number and costly with the collective financial impact of the infections reaching into the billions of dollars annually [23].

Treatments for HAIs increase the average length of stay for patients resulting in a reduction in hospital profit margins due to the increased number of days of uncompensated care which may significantly reduce hospital profit margins. However, what was not well established in the extant literature is which of HAI conditions have the most direct association with hospital profitability. Thus, our study aimed to explore the association between HAIs and hospital financial performance. Our primary findings indicated that C. *diff* and CAUTI rates were positively associated with an increase in operating expense per staffed bed, while increases in MRSA, C. *diff*, and CAUTI rates were correlated with increased operating expenses per adjusted discharge. In general, our findings agree with other studies that found significantly increased expenses associated with the treatment of HAIs [21,22,23].

As we consider our findings related to our first dependent variable, the operating expense per staffed bed, we interpret our results to mean that for every one percentage point increase in C. *diff* infection rates, we observed a 0.037% increase in operating expense per staffed bed, and for every one percentage point increase in CAUTI infection rates, we observed a 0.031% increase in operating expense per staffed bed. While these values appear to be small, from a practical standpoint, this means that for every 1% increase in C. *diff* infection rates, operating expense per bed increased by US $686 on average (based on a mean operating expense per bed of US $1.8 M). Similarly, CAUTI infections raise the operating expense per staffed bed by US $575 on average for each percentage point increase in this infection rate. When considered across the average size of the hospitals in our study (min: 50, max: 1498, mean: 285), the total cost per one percentage point increase in C. *diff* infections averaged US $195.5 K. Similarly, the total cost per one percentage point increase in CAUTI infections averaged US $163.8 K. Thus, if there is a sizable increase or decrease in the infection rate, the additional costs or cost savings on a per-bed basis can be substantial.

Second, our findings related to our second dependent variable the operating expense per adjusted discharge indicated that for every one percentage point increase in MRSA infection, operating expense per adjusted discharge increased by 0.042% while C. *diff* and CAUTI infections raised operating expenses by 0.062% and 0.039%, respectively. Thus, from a practical standpoint, based on an average operating expense per adjusted discharge of US $37.2 K, average per discharge costs increased by US $15.66, US $23.11, and US $14.54 related to MRSA, C. *diff*, and CAUTI with each one percentage point increase in infection rates, respectively. Once again, these values seem very small, until one considers that, on average, our sample facilities generated over 14,000 adjusted discharges in the year studied (min: 2080, max: 101,438, mean: 14,371). This means, on average, MRSA infections added US$219.2 K in operational expense per one percentage point increase in infection rates, while C. *diff* infections cost the facilities an added US $323.5 K, and CAUTI infections added US $203.5 K in operational expenses that were subtracted from the organization’s bottom line. Thus, an increase of as little as 2–3% in infection rates could easily drive excess operational costs into the millions of dollars. Conversely, appropriate attention and focus on these disease processes will inherently lead to cost avoidance that can now be captured with some accuracy based on our findings.

### Practice Implications

The practical implication of our research is that it provides healthcare leaders with some guidance and supportive evidence to assist in underwriting HAI reduction efforts and improve overall. To promote the financial benefit of HAI reduction, some states have incentives that encourage healthcare providers to sustain prevention and reduction efforts with subsidies for equipment, while the hospital’s Medicare reimbursement rates may increase for meeting their HAI reduction targets [33]. Additionally, the CDC will assist healthcare facilities with high infection rates that seek to establish preventive measures to reduce HAIs [34]

Our research results further support these national and state efforts by providing hospital leaders with some clarity related to which HAI infections have the most significant and meaningful association with hospital profitability. Arguably, administrators and clinical leaders can use our results to advance the cost-benefit argument in support of HAI reduction efforts. Numerous strategies to reduce HAI levels have been recommended by prior authors, but each comes with an inherent cost. For example, at the facility level, Gidey al. (2023) suggested that evidence-based preventative measures like infection control procedures could prevent or reduce HAI infection rates [32]. Revelas (2012) recommended encouraging hospital visitors to wash their hands and use gloves (as appropriate), and they noted that doctors, as compared to nurses, need to increase the frequency of their handwashing [5]. The CDC encourages hospital administrators to interface with clinical staff to establish a standing multidepartment task force to monitor the HAI infection rates, to proactively address HAI reduction measures by ensuring adequate clinical staff and infection control department staff levels, by aggressively monitoring compliance with infection control policies [35]. The authors assert that this committee could perform observational studies of handwashing hygiene with staff training on proper handwashing techniques, and they could support institutional efforts to provide environmental cleanliness (e.g., bedside tables, medicine carts, IV poles). These committee members and other health leaders could also interface with the CDC’s Healthcare-associated Infections and Antimicrobial Resistance (HAI/AR) Program which can help hospitals start or improve their antibiotic effort initiatives and they can help them partner with academic institutes to conduct studies of HAIs among their patient population [35].

Beyond these recommendations from prior authors, some practical financial benefits accrue from reducing HAIs that can reduce the total cost of care and improve revenue generating potential for hospitals. First, hospital beds, formerly occupied by HAI patients accumulating uncompensated days of care, can be released to patients eligible for third party payer reimbursement [36]. Second, reducing HAIs can reduce the costs associated with hiring additional staff, particularly additional higher cost contract nursing labor and other clinical staff, to cover the excess bed days and perform the extra diagnostic tests (lab, radiology) for these patients. Lastly, an intangible, but no less important, benefit to reducing the number of HAIs is that it can improve patients’, payers, and providers’ perceptions of the hospital which can translate into increased revenue in future reimbursement contract negotiations and willingness of providers to admit patients and perform surgeries in the facility.

We should also note that although HAIs are not directly reimbursed, the cost of HAI treatment inherently increases the total cost of care that hospitals must address financially in some way. While these costs cannot be passed on to government payers (e.g., Medicare, Medicaid, TRICARE, etc.), they can be included in the increased rates included on each hospital’s chargemaster and thus become an initial point of negotiation for any commercial insurer contract negotiations. As we follow this logic, this would mean that any individual or organization that pays for commercial insurance is ultimately paying for the cost of HAI treatment to some degree. In the past, this additional cost-shifting to private payers has been estimated to cost an additional $25 billion to $35 billion in acute care costs annually [37].

## 5. Limitations and Suggestions for Future Research

This study has strengths in the in-depth exploration of the association between HAIs and hospital financial performance using recent data. To our knowledge, this study is one of the few studies to explore the specific large set of predictors for a large national sample of US hospitals and four different nosocomial infections. As with all research, this study has some limitations.

First, we use data for the calendar year 2022, a longitudinal study may provide future researchers with additional understanding of the association between HAIs and financial performance in the hospital setting. A longitudinal approach might also assist a future research team to move beyond our cross-sectional non-experimental study design and guide future efforts to determine causality between HAIs and hospital financial performance.

Second, there may be additional predictors that affect hospital financial performance that were not included in our study, but which could reveal new insights into the HAI infection and hospital profitability connection. For example, individual patient demographics, clinical staff composition, and hospital service mix might all yield interesting results. Unfortunately, these variables were not readily available to the research team for our sample hospitals.

## 6. Conclusions

Maintaining profitability in the contemporary healthcare operating environment has never been more challenging for hospital leaders. With inflationary increases in medical supply costs and equipment coupled with challenges with maintaining adequate staffing, there is little room for error if economic viability is to be sustained. Our findings underscore the critical importance of proactively addressing healthcare acquired infections. By implementing targeted interventions to prevent HAIs, hospitals can not only improve patient outcomes from a clinical standpoint, but they can also significantly reduce the overall cost of care. This dual benefit is essential for sustaining hospital financial performance and ensuring long-term economic viability in an increasingly uncertain and demanding healthcare environment.

## Figures and Tables

**Table 1 healthcare-12-01314-t001:** Descriptive Statistics.

	Min	Max	Mean	Std Dev
**Dependent Variables**				
Operating Expense per Staffed Bed	387,815.70	8,159,529.41	1,855,671.61	896,342.27
Operating Expense per Adjusted Discharge	7150.29	192,629.67	37,280.97	16,644.75
**Independent Variables of Interest**				
Methicillin-resistant Staphylococcus aureus (MRSA) Infection Rate	0.00	4.35	0.82	0.66
Clostridium Difficile (C. diff) Infection Rate	0.00	2.18	0.44	0.27
Catheter-Associated Urinary Tract Infections (CAUTI) Rate	0.00	5.19	0.64	0.51
Central-Line-Associated Blood Stream Infections (CLABSI) Rate	0.00	4.84	0.75	0.61
**Control Variables**				
Contract Labor (in millions)	0.00	124.92	20.05	21.88
Managed/Leased Facility	0	1	0.04	0.19
Sole Community Hospital	0	1	0.08	0.27
Accountable Care Organization Affiliated	0	1	0.71	0.46
Labor Compensation Ratio	17.60%	119.00%	0.45	0.13
Market Concentration Index	0.02	1.00	0.28	0.29
Number of Employees	26	16,152	1960.60	1688.74
Government Operated	0	1	0.11	0.32
For Profit Hospital	0	1	0.17	0.38
Academic Medical Center	0	1	0.09	0.28
Rural Geographic Classification	0	1	0.08	0.27
Uncompensated Care (in millions)	0.01	673.44	34.80	50.65
Average Length of Stay	1.50	9.80	5.14	0.88
Payor Mix: Percent Medicare Days	0.10%	64.40%	0.26	0.09
Payor Mix: Percent Medicaid Days	0.10%	68.60%	0.08	0.08
Case Mix Index	1.28	3.37	1.85	0.27
Complication/Major Complication (CC/MCC) Rate	0.31	0.88	0.69	0.05
Hospital Value Based Purchasing Total Performance Score	2.50	56.75	20.63	8.15
Region 1 (CT, ME, MA, NH, RI, VT)	0	1	0.05	0.32
Region 2 (NJ, NY, PA)	0	1	0.14	0.34
Region 3 (DE, KY, MD, NC, VA, WV, DC)	0	1	0.09	0.28
Region 4 (AL, FL, GA, MS, SC, TN, PR)	0	1	0.17	0.38
Region 5 (IL, MI, IN, OH, WI)	0	1	0.16	0.37
Region 6 (IA, KS, MN, MO, NE, ND, SD)	0	1	0.07	0.25
Region 7 (AR, LA, OK, TX)	0	1	0.12	0.32
Region 8 (AZ, CO, ID, MT, NM, UT, WY)	0	1	0.07	0.25
Region 9 (AK, CA, HI, NV, OR, WA)	0	1	0.14	0.34

**Table 2 healthcare-12-01314-t002:** Regression Analysis Results—Operating Expense per Staffed Bed.

Analysis of Association between Healthcare Associated Infections & Hospital Financial Performance	Natural Log of Operating Expense per Staffed Bed											
	N = 1454, Adj R^2^ = 47.8%			N = 1454, Adj R^2^ = 48.1%			N = 1454, Adj R^2^ = 48.0%			N = 1454, Adj R^2^ = 47.8%		
	β	S.E.	Sig	β	S.E.	Sig	β	S.E.	Sig	β	S.E.	Sig
**INDEPENDENT VARIABLES**												
MRSA Infection Rate—Natural Log	0.004	0.016	-									
C. diff Infection Rate—Natural Log				0.037	0.014	*						
CAUTI Infection Rate—Natural Log							0.031	0.014	*			
CLABSI Infection Rate—Natural Log										−0.006	0.015	-
**FACILITY ATTRIBUTES**												
Contract Labor (in millions)	0.001	0.001	*	0.001	0.001	-	0.001	0.001	*	0.001	0.001	*
Managed/Leased Facility	−0.027	0.045	-	−0.034	0.045	-	−0.033	0.045	-	−0.025	0.045	-
Sole Community Hospital	0.035	0.040	-	0.034	0.039	-	0.033	0.039	-	0.034	0.040	-
Accountable Care Organization Affiliated	−0.009	0.020	-	−0.01	0.020	-	−0.009	0.020	-	−0.009	0.020	-
Labor Compensation Ratio	0.000	0.001	-	−0.001	0.001	-	0.000	0.001	-	0.000	0.001	-
Market Concentration Index	0.094	0.039	*	0.089	0.039	*	0.093	0.039	*	0.093	0.039	*
Number of Employees	0.001	0.000	***	0.001	0.000	***	0.001	0.000	***	0.001	0.000	***
Government Operated	0.050	0.031	-	0.044	0.031	-	0.045	0.031	-	0.050	0.031	-
For Profit Hospital	−0.299	0.026	***	−0.293	0.026	***	−0.299	0.026	***	−0.300	0.026	***
Academic Medical Center	0.034	0.037	-	0.031	0.037	-	0.029	0.037	-	0.036	0.037	-
Rural Geographic Classification	−0.010	0.038	-	−0.014	0.038	-	−0.008	0.038	-	−0.009	0.039	-
Uncompensated Care (in millions)	0.001	0.000	***	0.001	0.000	***	0.001	0.000	***	0.001	0.000	***
Average Length of Stay	−0.024	0.012	*	−0.025	0.012	*	−0.024	0.012	*	−0.024	0.012	*
Payor Mix: Percent Medicare Days	0.002	0.001	*	0.002	0.001	*	0.002	0.001	*	0.002	0.001	*
Payor Mix: Percent Medicaid Days	−0.003	0.001	*	−0.003	0.001	*	−0.003	0.001	*	−0.003	0.001	*
Case Mix Index	0.261	0.044	***	0.259	0.044	***	0.271	0.044	***	0.259	0.044	***
Complication/Major Complication (CC/MCC)	0.208	0.170	-	0.22	0.169	-	0.197	0.170	-	0.205	0.170	-
HVBP Total Performance Score	0.009	0.001	***	0.009	0.001	***	0.009	0.001	***	0.009	0.001	***
Region 2 (NJ, NY, PA)	−0.318	0.045	***	−0.316	0.045	***	−0.313	0.045	***	−0.317	0.045	***
Region 3 (DE, KY, MD, NC, VA, WV, DC)	−0.346	0.050	***	−0.337	0.050	***	−0.341	0.050	***	−0.344	0.050	***
Region 4 (AL, FL, GA, MS, SC, TN, PR)	−0.485	0.046	***	−0.475	0.045	***	−0.475	0.046	***	−0.483	0.045	***
Region 5 (IL, MI, IN, OH, WI)	−0.299	0.045	***	−0.302	0.045	***	−0.294	0.045	***	−0.297	0.045	***
Region 6 (IA, KS, MN, MO, NE, ND, SD)	−0.395	0.050	***	−0.397	0.05	***	−0.396	0.050	***	−0.394	0.050	***
Region 7 (AR, LA, OK, TX)	−0.457	0.048	***	−0.45	0.048	***	−0.448	0.048	***	−0.456	0.048	***
Region 8 (AZ, CO, ID, MT, NM, UT, WY)	−0.246	0.054	***	−0.251	0.054	***	−0.240	0.054	***	−0.246	0.054	***
Region 9 (AK, CA, HI, NV, OR, WA)	−0.144	0.047	**	−0.148	0.047	**	−0.143	0.047	**	−0.142	0.047	**

Note: *p* is not significant, * *p* < 0.05; ** *p* < 0.01; *** *p* < 0.001; Region 1 is the referent region for analysis purposes.

**Table 3 healthcare-12-01314-t003:** Regression Analysis Results—Operating Expense per Adjusted Discharge.

Analysis of Association between Healthcare Associated Infections & Hospital Financial Performance	Natural Log of Operating Expense per Adjusted Discharge											
	N = 1454, Adj R^2^ = 58.0%			N = 1454, Adj R^2^ = 58.6%			N = 1454, Adj R^2^ = 58.0%			N = 1454, Adj R^2^ = 57.7%		
	β	S.E.	Sig	β	S.E.	Sig	β	S.E.	Sig	β	S.E.	Sig
**INDEPENDENT VARIABLES**												
MRSA Infection Rate—Natural Log	0.042	0.013	***									
C. *diff* Infection Rate—Natural Log				0.062	0.011	***						
CAUTI Infection Rate—Natural Log							0.039	0.011	***			
CLABSI Infection Rate—Natural Log										0.010	0.012	-
**FACILITY ATTRIBUTES**												
Contract Labor (in millions)	0.000	0.000	-	0.000	0.000	-	0.000	0.000	-	0.000	0.000	-
Managed/Leased Facility	0.043	0.036	-	0.036	0.036	-	0.040	0.036	-	0.044	0.036	-
Sole Community Hospital	0.052	0.032	-	0.051	0.032	-	0.050	0.032	-	0.054	0.032	-
Accountable Care Organization Affiliated	−0.018	0.016	-	−0.023	0.016	-	−0.021	0.016	-	−0.020	0.016	-
Labor Compensation Ratio	0.003	0.001	***	0.003	0.001	***	0.003	0.001	***	0.003	0.001	***
Market Concentration Index	0.175	0.032	***	0.164	0.032	***	0.169	0.032	***	0.170	0.032	***
Number of Employees	0.001	0.000	***	0.001	0.000	**	0.001	0.000	***	0.001	0.000	**
Government Operated	0.089	0.025	***	0.081	0.025	***	0.084	0.025	***	0.090	0.025	***
For Profit Hospital	−0.280	0.021	***	−0.271	0.021	***	−0.280	0.021	***	−0.281	0.021	***
Academic Medical Center	0.083	0.030	**	0.085	0.029	**	0.083	0.030	**	0.087	0.030	**
Rural Geographic Classification	0.078	0.031	*	0.073	0.031	*	0.082	0.031	**	0.078	0.031	*
Uncompensated Care (in millions)	0.001	0.000	***	0.001	0.000	***	0.001	0.000	***	0.001	0.000	***
Average Length of Stay	0.105	0.010	***	0.103	0.009	***	0.105	0.010	***	0.105	0.010	***
Payor Mix: Percent Medicare Days	0.002	0.001	*	0.002	0.001	-	0.002	0.001	*	0.002	0.001	*
Payor Mix: Percent Medicaid Days	−0.003	0.001	**	−0.003	0.001	**	−0.003	0.001	**	−0.003	0.001	**
Case Mix Index	0.360	0.035	***	0.352	0.035	***	0.368	0.036	***	0.357	0.036	***
Complication/Major Complication (CC/MCC)	−0.049	0.137	-	−0.030	0.136	-	−0.065	0.137	-	−0.048	0.137	-
HVBP Total Performance Score	0.006	0.001	***	0.007	0.001	***	0.006	0.001	***	0.006	0.001	***
Region 2 (NJ, NY, PA)	−0.197	0.036	***	−0.188	0.036	***	−0.185	0.036	***	−0.191	0.036	***
Region 3 (DE, KY, MD, NC, VA, WV, DC)	−0.260	0.040	***	−0.238	0.040	***	−0.247	0.040	***	−0.255	0.040	***
Region 4 (AL, FL, GA, MS, SC, TN, PR)	−0.459	0.037	***	−0.432	0.036	***	−0.435	0.037	***	−0.448	0.037	***
Region 5 (IL, MI, IN, OH, WI)	−0.141	0.036	***	−0.143	0.036	***	−0.132	0.036	***	−0.138	0.036	***
Region 6 (IA, KS, MN, MO, NE, ND, SD)	−0.212	0.040	***	−0.210	0.040	***	−0.208	0.040	***	−0.207	0.041	***
Region 7 (AR, LA, OK, TX)	−0.366	0.039	***	−0.348	0.039	***	−0.347	0.039	***	−0.360	0.039	***
Region 8 (AZ, CO, ID, MT, NM, UT, WY)	−0.148	0.043	***	−0.151	0.043	***	−0.136	0.043	**	−0.143	0.043	***
Region 9 (AK, CA, HI, NV, OR, WA)	−0.040	0.038	-	−0.037	0.037	-	−0.030	0.038	-	−0.031	0.038	-

Note: *p* is not significant; * *p* < 0.05; ** *p* < 0.01; *** *p* < 0.001; Region 1 is the referent region for analysis purposes.

## Data Availability

All analyses were conducted in SPSS, Version 28, and all tables were constructed in Microsoft Excel 2019.

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
