# Peer review of "An Exploratory Analysis of the Association between Healthcare Associated Infections & Hospital Financial Performance"

_healthcare, 2024, doi:10.3390/healthcare12131314_

Round 1

Reviewer 1 Report

Comments and Suggestions for Authors

The article meets the requirements for scientific studies. The research methods used allowed to achieve the research goal. The authors should supplement the discussion of the results with publications covering the research topic. In section 3.1, the purpose of the research should be clearly defined. The statement "We hypothesize that each of our targeted variables is statistically significantly associated with organizational costs even in the presence of numerous cost-driving organizational control variables" requires explanation. Is this a research hypothesis?

Author Response

REVIEWER 1

Comment 1: The article meets the requirements for scientific studies. The research methods used allowed to achieve the research goal.

Response 1: Thank you for your comment. We appreciate your feedback!

##################################################################
Comment 2: The authors should supplement the discussion of the results with publications covering the research topic.

Response 2: Thank you for this suggestion. We have re-written the discussion and included several publications related to HAIs. ##################################################################
Comment 3: In section 3.1, the purpose of the research should be clearly defined. The statement "We hypothesize that each of our targeted variables is statistically significantly associated with organizational costs even in the presence of numerous cost-driving organizational control variables" requires explanation. Is this a research hypothesis?

Response 3: In this revision, a refined research hypothesis is formulated and articulated and added to the Materials and Methods section.

Reviewer 2 Report

Comments and Suggestions for Authors

General comments:

The abstract states “A better understanding of which HAIs are more costly can provide guidance to hospital administrators for targeted interventions to reduce HAI prevalence.” How did this study contribute to a better understanding of which HAIs are costlier?  What impact does its results have on providing guidance to hospital administrators for targeted interventions to reduce HAI prevalence?

From the title and abstract I was hoping to see an analysis that revealed a way to provide improved profitability to medical facilities that reduced unreimbursed HAIs. One that looked at the frequencies of each HAI in relationship to its cost of treatment. For example, if higher cost HAIs of low frequency were less expensive to treat this would suggest greater emphasis on lower cost HAIs of high frequency.

Several other questions remain from the Conclusion section. How do hospital administrators interface better with medical staff providers to proactively address HAIs? Are any incentives offered for decreasing the incidence of HAIs (e.g., purchasing new diagnostic equipment, enhanced medical procedure training, etc.)? What targeted interventions are suggested by the study results to help prevent HAIs?

All bacterial names should be italicized and the first letter of the genus needs to be uppercase.

Specific Comments:

Line 15 - Suggest adding “third-party” in front of “payer”.

Line 19 - Recommend changing “more costly” to “costlier”.

Lines 24-25 - Change “Methicillin Resistant Staphylococcus Aureus” to methicillin-resistant Staphylococcus aureus. Similarly, “C. diff” should be designated as C. diff.

Lines 60-61- The criteria for classifying an infection as a HAI (e.g., 48 hour after admission) rather than a community-acquired infection (CAI) should be stated and referenced.

Lines 61-62 - Change “Center for Disease  Control” to “Center for Disease Control and Prevention”.

Line 71 - Define sepsis.

Line 77 - Suggest using “changes” rather than “maintenance”.

Line 79 - Suggest using “microbes’ instead of “germs”. Same comment for line 81.

Lines 83-84 - Replaceperson to  person” with “person-to-person”.

Line 84 - Replace “unclean” with “contaminated’.

Line 102 - Place brackets around reference number “18”.

Line 131 - Why did the authors decide upon using “short-term acute care hospital data”? Was it because these hospitals are the most common type in the US? Why were long-term care facilities (LTC) excluded? HAIs data derived from LTC facilities may be markedly different.

Line 181 - Why wasn’t ventilator-associated pneumonia (VAP) also included in the study as an independent variable of interest? It was listed as a common HAI in the Introduction on lines 64 and 105.

Line 315 - Change “HAIS” to HAIs”.

Comments on the Quality of English Language

Quality of English was fine w/ noted exceptions.

Author Response

REVIEWER 2

Comment 4: The abstract states “A better understanding of which HAIs are more costly can provide guidance to hospital administrators for targeted interventions to reduce HAI prevalence.” How did this study contribute to a better understanding of which HAIs are costlier?  What impact does its results have on providing guidance to hospital administrators for targeted interventions to reduce HAI prevalence?

Response 4: Thank you for your comment. In lines 299 – 330 of the updated draft, we note our statistically significant findings related to the cost of our studied HAIs. We hope this sufficiently addresses your concern.

##################################################################
Comment 5: From the title and abstract I was hoping to see an analysis that revealed a way to provide improved profitability to medical facilities that reduced unreimbursed HAIs. One that looked at the frequencies of each HAI in relationship to its cost of treatment. For example, if higher cost HAIs of low frequency were less expensive to treat this would suggest greater emphasis on lower cost HAIs of high frequency.

Response 5: Thank you for this comment. As we consider our results in lines 299 – 330, we can identify the cost per HAI within our study sample. This can assist hospital leaders in establishing the benefit (cost avoidance) in an HAI reduction effort. However, given the variance in direct and indirect interventions and treatments for these conditions that may be pursued by hospital leaders, we cannot reasonably add clarity on the cost of intervention to ameliorate each of our studied HAI conditions.  

##################################################################
Comment 6: Several other questions remain from the Conclusion section. How do hospital administrators interface better with medical staff providers to proactively address HAIs? Are any incentives offered for decreasing the incidence of HAIs (e.g., purchasing new diagnostic equipment, enhanced medical procedure training, etc.)? What targeted interventions are suggested by the study results to help prevent HAIs?

Response 6: Thank you for this comment. Although the specifics related to treatment and intervention of HAIs is beyond the original scope of our paper, we updated the Discussion and Practical Implications section to address national, state, and prior authors’ suggestions pertaining to HAI prevention. We hope these additions adequately address your concerns.

 ##################################################################
Comment 7: All bacterial names should be italicized and the first letter of the genus needs to be uppercase.

Response 7: We have made this change throughout the paper.

##################################################################
Comment 8: Line 15 - Suggest adding “third-party” in front of “payer”.

Response 8: We added “third-party” in front of “payer”.

 ##################################################################
Comment 9: Line 19 - Recommend changing “more costly” to “costlier”.

Response 9: We changed “more costly” to “costlier”.
##################################################################
Comment 10: Lines 24-25 - Change “Methicillin Resistant Staphylococcus Aureus” to methicillin-resistant Staphylococcus aureus. Similarly, “C. diff” should be designated as C. diff.

 Response 10: “Methicillin Resistant Staphylococcus Aureus” was changed to methicillin-resistant Staphylococcus aureus and “C. Diff” to C. diff.

 ##################################################################
Comment 11: Lines 60-61- The criteria for classifying an infection as a HAI (e.g., 48 hour after admission) rather than a community-acquired infection (CAI) should be stated and referenced.

Response: 11: The criteria for classifying an infection as a HAI (e.g., 48 hours after admission) has been added to the paper and referenced

##################################################################
Comment 12: Lines 61-62 - Change “Center for Disease Control” to “Center for Disease Control and Prevention”.

Response 12: We changed “Center for Disease Control” to “Center for Disease Control and Prevention”.

##################################################################

Comment 13: Line 71 - Define sepsis.

Response 13: Sepsis has been defined.

##################################################################

Comment 14: Line 77 - Suggest using “changes” rather than “maintenance”.

Response 14: We replaced “changes” with “maintenance”

##################################################################

Comment 15: Line 79 - Suggest using “microbes’ instead of “germs”. Same comment for line 81.

 Response 15: We changed microbes to germs.

##################################################################

Comment 16: Lines 83-84 - Replace “person to person” with “person-to-person”.

Response 16: Person to person was replaced with “person-to-person”

##################################################################

Comment 17: Line 84 - Replace “unclean” with “contaminated’.

Response 17: Replaced “unclean” with “contaminated’.

##################################################################

Comment 18: Line 102 - Place brackets around reference number “18”. ???

Response 18: The references have been updated in the revised draft.

##################################################################

Comment 19: Line 131 - Why did the authors decide upon using “short-term acute care hospital data”? Was it because these hospitals are the most common type in the US? Why were long-term care facilities (LTC) excluded? HAIs data derived from LTC facilities may be markedly different.

Response 19: Long-term care facilities are significantly different in numerous ways when compared to short term acute care hospitals in the United States. We agree that HAI data derived from LTC facilities and the association with organizational financial outcomes may be markedly different than what we discovered. However, this is a great suggestion for future researchers to consider and advance our findings.

##################################################################

Comment 20: Line 181 - Why wasn’t ventilator-associated pneumonia (VAP) also included in the study as an independent variable of interest? It was listed as a common HAI in the Introduction on lines 64 and 105.

Response 20: Unfortunately, a sufficient number of data points for VAP was not available in our database.

##################################################################

Comment 21: Line 315 - Change “HAIS” to HAIs”.

Response 21: We changed “HAIS” to HAIs”

Reviewer 3 Report

Comments and Suggestions for Authors

N/A

Author Response

Thank you for your review. We observed no comments to which we needed to review and respond.

Reviewer 4 Report

Comments and Suggestions for Authors

The authors have aaddressed an important topic. 

I have some querries and suggestions as mentioned below.

1. line 44 "with nearly 80 Chapter" what do you mean by chapters here?

2. The authors have focused on the healthcare system and financial aspects that are strictly for the US hospitals, in my opinion it will be better to describe in the begining how the healthcare system in US is funded, as it seems to be entirely different from many countries. This explanation will allow the reader to get a better insight of the problem.

3. HAI are indeed of major concern always, but how is it relevant whether it is a pre-covid or post -covid period?

4. line 102 what is "18" if it is a reference, kindly put it in order

5. all over the manuscript I would suggest adding US to the $ sign, for clarity

6. line 148-151, can you explain or rephrase it

7. all the tables should have the details of the abbreviations used in the legend sections

8. Since the article focuses on a one country problem, can the authors highlight the benefits for the rest of the world in planning cost-control. by rewritting the conclusion and future stratergies. 

Comments on the Quality of English Language

English is fine, just minor grammatical errors should be addressed and punctuations. 

Author Response

REVIEWER 3

Comment 22: The authors have addressed an important topic. I have some queries and suggestions as mentioned below.

  1. line 44 "with nearly 80 Chapter" what do you mean by chapters here?

Response 22: Thank you for your comment. We added the work bankruptcy to clarify that Chapter 11 filings are for bankruptcies.

##################################################################

Comment 23: 2. The authors have focused on the healthcare system and financial aspects that are strictly for the US hospitals, in my opinion it will be better to describe in the beginning how the healthcare system in US is funded, as it seems to be entirely different from many countries. This explanation will allow the reader to get a better insight of the problem.

Response 23: We have added a brief explanation of how US short term acute care hospitals are funded in the Introduction section of the revised draft.

##################################################################

Comment 24: 3. HAI are indeed of major concern always, but how is it relevant whether it is a pre-covid or post -covid period?

Response 24: We have removed the reference to the Covid period. We have analyzed data for one calendar year, the year 2022, thus due to data limitations we cannot comment on pre- and post- Covid periods.

##################################################################

Comment 25: 4. line 102 what is "18" if it is a reference, kindly put it in order

Response 25: The references have been updated.

##################################################################

Comment 26:  5. all over the manuscript I would suggest adding US to the $ sign, for clarity

Response 26: The US was added to the $ sign for clarity.

##################################################################

Comment 27:  6. line 148-151, can you explain or rephrase it

Response 27: These lines have been revised for clarity.

##################################################################

Comment 28: 7. all the tables should have the details of the abbreviations used in the legend sections

Response 28: The tables have been revised and abbreviations eliminated, where possible.

##################################################################

Comment 29: 8. Since the article focuses on a one country problem, can the authors highlight the benefits for the rest of the world in planning cost-control. by rewriting the conclusion and future strategies.

Response 29: The conclusion and future strategies have been revised to highlight the benefits for the rest of the world in planning cost-control.

Round 2

Reviewer 2 Report

Comments and Suggestions for Authors

Author responses to my comments are sufficient. No further comments.